# Value Generation of Remanufactured Products: Multi-Case Study of Third-Party Companies

**Fernando Tobal Berssaneti** [1,*] , **Simone Berger** [1], **Ana Maria Saut** [1] ,
**Rosangela Maria Vanalle** [2] **and José Carlos Curvelo Santana** [2]

1. Departamento de Engenharia de Produção, Escola Politécnica da Universidade de São Paulo, São Paulo 05508-010, Brazil; siberger@usp.br (S.B.); ana.saut@usp.br (A.M.S.)
2. Programa de Pós-Graduação em Engenharia de Produção, Universidade Nove de Julho; São Paulo 03155-000, Brazil; rvanalle@uni9.pro.br (R.M.V.); jccurvelo@uni9.pro.br or jccurvelo@yahoo.com.br (J.C.C.S.)
* Correspondence: fernando.berssaneti@usp.br; Tel.: +55-11-5525-8742

**Abstract:** The importance of reverse logistics has increased owing to environmental factors and recent legislations. In this context, the market for remanufactured goods has become attractive. Manufacturers, retailers, and third-party companies have improved return programs and operations that add value to the return chain for electronic appliances, rather than treating it as a secondary process. The objective of this study is to identify the variables related to value generation in the reverse logistics of electronic products from the perspective of third-party companies. Reverse logistics of electronic products depends much on the context and local regulations; in addition, the fact that there are few studies on developing countries points to an important gap in extant research. This study presents the influence of quality and warranties, processing time, and partnerships between third-party companies, manufacturers, and retailers on the value generation from remanufactured products. These variables are related to optimal results and optimistic expectations for growth among third-party companies. These internal factors, together with an analysis of external factors and product portfolios, complement the scenario description for the cases studied. The main contribution of this study is to highlight the major factors, which are presented in the framework. The lessons learned can be used in other contexts involving third-party companies.

**Keywords:** electronic appliance; reverse logistics; partnership; refurbished; remanufactured

## 1. Introduction

The number of electrical and electronic equipment (EEE) has increased owing to the development of information technologies in the last 30 years. About 50 million tons of waste equipment is estimated to have been generated worldwide in 2015 [1,2]; a similar volume is also expected in 2018 [3,4].

Rapid advancements in material science, manufacturing processes, and electronic products have created global markets that are witness to rapid diffusion of technology to consumers. In recent years, advancements in telecommunication and information technologies have accelerated with globalisation, making it possible to develop markets for the new products on scales that are larger than were possible earlier in terms of data acquisition, product dissemination, technology application, consumer behaviour, and market penetration. The increase in the production of consumer products and their global distribution, combined with their affordability, have created challenges for managing municipal solid waste, especially the growing quantities of discarded electronic consumer products [5].

This has led to an increase in solid waste generation worldwide. Consequently, discussions about the environmental problems caused by the waste electrical and electronic equipment (WEEE) have sought to address the issue of waste being discarded without a minimum level of treatment. Even in

developed countries, there are reports of irregular discarding of electronic waste, such as microwave disposal in the UK, by consumers [6,7].

Milovantseva and Fitzpatrick [1] noted that improper disposal or crude processing of discarded WEEE may cause hazardous materials to contaminate soil, water, and air through various processes and affect human health. Pascale et al. found that discarding electronic devices significantly contributes to environmental pollution. Harm to human health has also been documented among workers and other vulnerable groups.

The process of returning used products and components for re-use in the supply chain is known as reverse logistics. The term refers to the logistics activities inside an organization that move in a direction opposite to that of normal supply chain activities [8]. Reverse Logistics includes all activities associated with product recovery such as repairing, recycling, remanufacturing, and disposing. The application of reverse logistics (RL) is expanded prominently due to environmental issues, and profit related to the returned products [9].

Reverse logistics has attracted the attention of many manufacturers because it is related to the recovery process of end-of-life products [10], which are collected from customers through collection centres and are assigned either for remanufacturing, recycling, or green disposal [11]. Many firms have started to apply integrated environmental strategies to improve their business models and competitive advantage through the reuse of, and recovery process for, end-of-life products [12].

Remanufacturing has gained an increasing amount of attention due to its contribution to energy saving and pollution reduction. However, remanufactured products have low customer acceptance because people are concerned about quality and after-sales service [9].

Unlike consumer durable goods, high technology products have shorter life cycles. Returned products can be restored using the remanufacturing process and offered to other customers. Therefore, multiple generations of new products and remanufactured products can be available side-by-side in the marketplace. New products are affected by technology obsolescence, while the remanufactured products are affected by quality and technology obsolescence [13].

Guide [14] addressed the importance of remanufacturing for the environmental and economic goals of companies by studying the difficulties involved with this process; these include the lack of structured management procedures in operations and logistics. Tibben-Lembke and Rogers [15] highlighted the difficulty of reverse logistics in demand planning, transportation, ensuring product availability to meet demand, pricing, speed of returns, reprocessing and sales, cost, and market conditions. Because manufacturers may consider remanufactured products a threat, there are few marketing and awareness campaigns.

The remanufactured electronics market has a yearly value of approximately USD 10 billion in the US alone [16]. An increase in e-commerce and flexibility in return policies have contributed to its growth. According to [17], a US investment company that analyses internet and technology trends, e-commerce sales in the US represented approximately 11% of all sales in 2015 and have been increasing at an annual rate of about 15%. Meanwhile, returns accounted for about 8% of all e-commerce sales. When the exchange of cheaper products is accounted for, the total value rises to 20% [18].

In Brazil, e-commerce has also been growing, representing about 13% of the total sales in the retail sector. This figure increased by 12% in 2017 [19]. According to the Programa de Proteção e Defesa do Consumidor (PROCON), consumers who buy products either online or over the phone can return them within a period of seven days. These products can be returned even if there is no defect; the client's regret over the purchase is sufficient reason for return.

The electronics market is also unusual in that it features the constant launch of products with new technologies and functionalities, thereby shortening the useful life of a given product and creating a market for second-hand goods, with functional equipment, that are sold for reuse [20]. The greater consumption of electronic appliances and the shortening of their useful life have accelerated product obsolescence. Millions of tons of products are discarded every year, but a large proportion of them are reusable [21].

In this context, the market for remanufactured goods has become attractive. Manufacturers, retailers, and third-party companies have improved return programs and operations that add value to the return chain for electronic appliances, rather than treating it as a secondary process. This is in line with concepts developed by Guide [14]. Product returns and reverse supply chains present an opportunity for generating value, rather than a financial loss [22].

Several studies on WEEE in various countries, including Brazil, have been conducted. However, none addresses the Brazilian remanufacturing industry, which is responsible for increasing the life cycle of EEE in Brazil. Thus, the objective of this article is to identify the variables related to value generation in the reverse logistics of electronic products from the perspective of third-party companies. Reverse logistics of electronic products depend on the context and local regulations. In addition, there are few studies on developing countries; thus, there exists an important research gap.

## 2. Literature Overview

### 2.1. Normatization of Waste Electrical and Electronic Equipment

Legislation has been developed to minimize the impact caused by WEEE and make its use more sustainable. European legislation primarily refers to Directive 2002/96/EC of the European Parliament and of the Council of 27 January 2003 and Directive 2012/19/EU of the European Parliament and of the Council of 4 July 2012. In addition, EU legislation includes the RoHS Directive 2002/95/EC, which contains restrictions on the use of hazardous substances in electrical and electronic equipment and has been in force since February 2003. Under Directive 2002/96/EC, the current collection rate is 4 kg/inhabitant and, starting in 2019, the minimum collection rate is required to be 65% of the average weight of the EEE introduced in the market [23].

According to Popescu [23], EU countries such as Norway, Sweden, Finland, Ireland, Germany, Belgium, Luxembourg and Austria, have recorded higher annual WEEE collection rates (over 8 kg per capita) since this directive was issued; however, most EU member countries did not meet the target. An example is Romania, which collected just 1.1 kg per capita.

Angouria-Tsorochidou et al. [24] examined Denmark's recycling system for electronic waste and found that it increased the reuse rate from less than 1% to 45%. All the equipment, supplies, utilities, and construction costs involved in the process were recovered within two years of its launch. The profit derived from the electronic products for reuse and recycling was EUR 650,000/year, showing that garbage can be turned into profit.

In 2009, the Council of Australian Governments (COAG) agreed that television and computer waste should be the main products to undergo extended producer responsibility regulations under the national Product Stewardship Scheme. After it came into force, a COAG analysis showed that industry-funded collection and recycling schemes for televisions and computers would provide societal benefits up to 2030–31 [25]. Hence, Australia sought to reach a target of 50% WEEE recycling by 2015–16 [25].

Some countries, such as India, have not yet implemented electronic waste management and recycling systems; therefore, WEEE is collected, along with other solid waste, and deposited in landfills [26].

Although it is one of the largest generators of electronic products in the world, China currently only has 25 enterprises that remanufacture electrical and electronic waste. They are supervised by the Chinese authorities and requested to provide balanced reports regarding quantity and product destination [27].

On 5 August 2010, Federal Law No. 12305, referring to the National Policy on Solid Waste in Brazil, was approved; it requires adequate disposal of solid waste, including electronic waste. Brazil generated 1.6 million tons of electronic waste in 2016, making it second only to the US in North and South America, which produced 6.3 million tons annually. In 2013, the Brazilian government published a sectoral agreement to improve the reverse logistics system for electronic products and

their components. However, electronic technical assistance was not included in this agreement. This agreement aimed to recycle 17% of the WEEE by 2018 [28,29].

## 2.2. Impact of Electrical and Electronic Equipment

As previously mentioned, the improper disposal or crude processing of discarded WEEE may cause hazardous materials to contaminate the soil, water, and air [1].

For example, according to Krause and Townsend [30], the lead concentration of electronic cigarettes exceeded the regulatory threshold by a factor of 1.5–10. Pérez-Belis et al. [31] stated this was similar to concentrations found in other e-waste, such as remote controls and toys.

According to Scruggs et al. [32], personal computers and mobile phones can contain over 1000 different substances, including valuable metals as well as hazardous substances (e.g., mercury, lead, cadmium, beryllium, hexavalent chromium, antimony, brominated flame-retardants, polyvinyl chloride, and polychlorinated biphenyls).

In addition, Cai et al. [33] showed, on the basis of the benzo/pyrene equivalent concentration, that the emissions of polycyclic aromatic hydrocarbons from electronic waste may have much stronger toxicity than if these were present in the gaseous phase.

According to Shahata and Trupp [34], the greater part of heavy metal cations are ecological poisons that have toxic effects on all living organisms. Among them, mercury is considered to be one of the most hazardous metal ions and carries toxic risks for the human environment, as it can be dispersed in water, air, and soil. It can cause developmental delays and health problems, including leukaemia and damage to the brain, nervous system, kidneys, and endocrine system.

As previously mentioned, Pascale et al. [7] warned that discarded electronic devices are significant contributors to environmental pollution. Furthermore, they documented the harm to human health among workers and other vulnerable groups, including children and pregnant women who may come in direct contact with toxic substances or experience airborne exposure.

## 2.3. WEEE Recycling Barriers

In the extended producer responsibility context, countries and trading blocs (e.g., EU WEEE Directive) were observed to place regulatory obligations on producers, importers, and retailers to cover lifecycle WEEE costs, thereby applying greater pressure to produce eco-sensitive designs and augment cleaner production.

However, according to Peng et al. [35], most enterprises have expectedly demonstrated low interest in extended producer responsibility systems owing to a wide range of barriers, including high cost, low efficiency on subsidy audits, a lack of effective and efficient collection systems, and low levels of public awareness and participation.

According to Bernard [36], the European model, which is regulated by the EU's Directive on Waste Electrical and Electronic Equipment, transfers the effects of pollution from rich countries to developing countries. Meanwhile, according to Lepawsky et al. [37] and Wang et al. [38], 'bad faith' is often used to solve the problems of electronic waste; it is also referred to as the 'Best-of-2-Worlds' (Bo2W) philosophy. This philosophy appears in the promiscuous relationship between developed and underdeveloped countries, which, in the case of electronic waste plays out in the following ways. The first scenario entails collecting discarded electronics in low-income countries, followed by manual disassembly in a low-income locale and subsequent export of high-purity metal fractions for refining in high-income countries [37,38]. The second scenario combines the first scenario with collection of discarded electronics in high-income countries, followed by export for manual disassembly in a low-income locale; subsequently, there is re-export of high-quality metal fractions for refining in high-income countries [37].

In Europe, according to Nowakowski [2], contamination of raw material incurred through shredding and separation using other materials or hazardous substances can reduce the price of raw materials from 20–40% depending on the contamination level. In addition, recycling

WEEE is associated with manual labour, which promotes the practice of sending these products to underdeveloped countries, where low-cost labour is prevalent [38].

According to Scruggs et al. [32], in many developing countries, particularly low- and middle-income ones, a significant proportion of WEEE components is sent to unsanitary (uncontrolled) landfill sites. Similarly, informal e-waste recycling is widely practiced. Wires are burned in open spaces to remove plastic and recover copper. Acid extraction is also practiced to retrieve precious metals, such as gold, platinum, palladium and silver from printed circuit boards (PCBs). Such practices can be found in countries such as China, India, Pakistan, Vietnam, the Philippines, Nigeria, and Ghana, where e-waste is disassembled using rudimentary methods to recover valuable metals by people who lack the facilities to safeguard the environment and public health [32,38,39]. In this way, workers in the waste recycling sector are exposed to various diseases caused by toxic substances in electrical and electronic waste.

The Brazilian remanufacturing industry has little interest in precious metals from WEEE; the remanufacturers are only interested in the reuse of parts and electronic components. This is because Brazilian environmental agencies have not yet implemented an environmental control policy that assesses the contamination of the environment by hazardous components of electrical and electronic waste [28].

Control measures within countries and worldwide could mitigate the problems caused by Bo2W practices. According to Lee et al. [40], many nations seek to control or prevent the inflow of WEEE but such flows are difficult to track due to undocumented, often illegal global trade in electrical and electronic waste. In fact, the control of waste generated and treated from electrical and electronic devices is difficult even in countries that have followed more stringent directives for more than 20 years, such as the countries of the European Union. Thus, the use of technology, such as the wireless GPS location trackers, QRCode, big data, and Internet of things, has become popular in recent years [2,4,40]; according to Ikhlayel [41], this practice can bring benefits in the short, medium, and long terms.

## 2.4. Viability of WEEE Recycling Techniques

Among the barriers, cost is cited most often by companies. However, according to Angouria-Tsorochidou et al. [24], the world production of mobile phones requires approximately 44 tons of gold, 455 tons of silver, and 16,381 tons of copper, all of which are very valuable.

Several studies have been carried out to show that the recycling process of WEEE materials is economically and environmentally viable [41]. Scruggs et al. [32] used mobile phones as an example; a ton of discarded phones can contain 150 g or more of gold (USD 40.81/g) and 10% and 0.3% by weight of copper (USD 7.03/kg) and silver (USD 0.36/g), respectively. Thus, the recycling of these mobile phone metals generates about USD 8000/t; this makes the process financially viable.

To recover metals from WEEE, pyrometallurgical, hydrometallurgical, and biohydrometallurgical techniques have been used [42,43]. Using an integrated process for recycling copper anode slime from electronic waste smelting, Dutta et al. [44] were able to recover gold, lead, tin, and antimony, with salvage rates of 98.1%, 99.1%, 92.3%, and 97.1%, respectively. Lee and Mishra [45] were able to recover 90% of iron and 98% of copper in electrical and electronic waste that was pre-treated by using mechanical and thermal methods. Zhang et al. [46] developed a more sustainable method of copper recovery from waste printed circuit boards using ionic liquids, and achieved a yield with 98% purity. Another sustainable way to recover precious metals from electronic waste that avoids chemicals is the use of microorganisms, such as *Escherichia coli* for copper recovery, which had a 95.2% extraction yield [47], and *Lactobacillus acidophilus* for gold recovery, with an 85% extraction yield [48].

On the other hand, Nowakowski [2] asserted that WEEE at the disassembly stage can change the potential revenue of raw materials. For example, the price offered by recycling companies in Poland for personal computer main boards is EUR 5–7/kg; that for PCB from hard disks is EUR 9–11/kg; boards from CD or DVD drives fetch EUR 2–3.5/kg; and mobile phone main boards are valued at EUR 21–23/kg.

In a case study of Brazilian WEEE recycling industries, Oliveira Neto and Correia [29] showed that for every 1000 tons of recycling, it is possible to earn a profit of about USD 3.2 million, with companies recouping their investment in less than three years. The authors cite gains from the extraction and sale of precious and semi-precious metals, plastics, glass and other materials, as well as benefits to the environment and human health. Among the other materials that are extracted, the literature cites rare elements, such as yttrium, gallium, tin, and germanium, as well as silicon, iron, zinc, and copper [49–52].

In manufacturing EEE, a variety of materials, such as ferrous and non-ferrous metals, plastics, precious metals, and rare earth metals, are applied; this makes possible the reuse of these materials after their dismantling. The designers take into consideration the last stage of the lifecycle in disassembling and recycling the product. Therefore, the networks between industries and remanufacturing companies must efficiently function for WEEE recycling processes to be viable [24].

Because production is the dominant contributor to EEE's total environmental impact, using phones—with their constant consumption of energy and natural resources—longer would be beneficial. This would hold even under rapid improvements in material and energy efficiency, as it has been shown that smartphones reduce their carbon footprint by about 30% just by extending their usage one year [53]. Thus, remanufacturing increases the lifecycle of the EEE and reduces its impact and, at the end of its cycle, the extraction of minerals improves the economic efficiency of these products.

## 2.5. Remanufactured Products

Green logistics is a form of logistics that includes environmental protection and sometimes even social issues, such as socioeconomic criteria [54]. From this perspective, reverse logistics is germane to the green logistics literature as a tool for improving the level of recovery and return of the used products; it can reduce pollution and product waste through activities such as refurbishing, repairing, remanufacturing, recycling, disposal, and parts recovery. Thus, it is necessary to use reverse logistics to find a way to increase the proportion of used products returned to the plant, thereby protecting the environment [11].

Reverse logistics is the key to green supply chain management. It helps firms achieve better performance and can be an opportunity to increase the rate of return on end-of-life products. These end-of-life products are remanufactured and recycled, or disposed of; because they can be used as raw materials or revived as products, it ultimately lead to gains for customers [11,55].

Lund [56] defined 'remanufacturing operations' as the process of collecting and recovering returned products.

Remanufacturing reduces the negative impact of waste products on the environment. In recent years, more and more manufacturers have implemented closed-loop operations and incorporated remanufacturing into their manufacturing system, thereby creating manufacturing-remanufacturing operations. Moreover, some electronic companies that produce disposable cameras, printers, ink cartridges, and copiers have similar action plans. Remanufacturing has gradually become a key business operation that electronic manufacturers should develop and promote. Apple, Samsung, and Xerox have also set up special remanufacturing facilities or subsidiaries to handle remanufacturing. In addition, these companies have been able to achieve operational goals related to energy conservation, business development, and profitability [57].

To be sold with the functionality of a new product, these products must meet manufacturing guidelines. Guide [14] described remanufacturing as a way to avoid waste through product reuse and value-add. This is an alternative to the material recovery obtained from recycling processes. According to Vorasayan and Ryan [58], remanufacturing must comply with high standards of quality, paying close attention to both the interior and exterior of the product. Electronic products undergo rigorous tests to ensure that they meet the original specifications of the manufacturer.

In the electronic industry, large numbers of electronic manufacturers have begun to recycle in recent years; however, their business models of used product collection are quite different. Some

enterprises, such as Samsung and Huawei, have set a fixed collection price for used products and collect them without consideration of the products' quality; however, some other enterprises choose to set a quality-dependent collection price for used products [57].

Blackburn et al. [22] described commercial product returns as cases in which a product is returned within a period of 90 days after its sale; manufacturers or retailers must find the most lucrative way of managing and reselling them. Reverse logistics consists of designing, planning, and controlling the collection, remanufacturing, value recovery, and sale of the returned products.

Various terms are used to describe remanufactured products, such as reconditioned, repackaged, and recertified. We use the definition provided by Abbey et al. [59], whereby the term 'remanufactured' refers to any product that underwent a process in which its appearance and functionality were conditioned for the sake of resale.

Dias et al. [60] studied the concept of sustainability in supply chains, wherein reverse logistics came to represent a potential competitive advantage for companies involved in a particular chain. Santos and Alves [61] considered sustainability in their model to integrate household appliance supply chain members, reinforcing their importance in the competition with other supply chains. Sustainability outcomes encompass the adoption of environmentally and socially responsible practices, as well as the achievement of environmental, social, or economic performance. Environmental practices include investments in pollution control and prevention, adoption of environmental management systems, and attainment of environmental certifications, such as ISO 14001 [62,63].

Mutha et al. [64] reinforced the need for agreed-upon policies to mediate third-party companies' relationships with manufacturers and retailers. Such agreements could establish partnerships for improving return, remanufacture, and, finally, sale in the remanufactured product market.

The literature about variables regarding the value generation in remanufactured electronic products can be grouped into quality and warranty, processing time, and partnership with manufacturers and retailers.

### 2.5.1. Quality and Warranty

Zikopoulos and Tagaras [65] explained how the impact of uncertainty on product quality affects the profitability of remanufacturing. The quality of returned products is associated with their appearance and/or functionality. The negative image of remanufactured products can be related to the expectation of external defects or issues about their functionality. External defects can be the result of mishandling during transportation and storage and require information and consumer awareness. On the other hand, failures in functionality demand certification and categorisation as 'new', 'certified repackaged', or 'remanufactured', according to the cause of return. In this sense, partnerships with retailers are necessary for facilitating return and classification, as well as treating returned products as sources of value, rather than waste [59].

Products under warranty can be returned in any physical state; after purchase, they can be returned after 30, 60, or 90 days of use [66]. In order to maximise financial returns, return chain planning requires analysis according to the product's life cycle stage. Depending on the quality of the product and the manufacturer's policies, they can be repackaged, remanufactured, or disassembled) [58].

Product quality affects remanufacturing costs. Third-party companies seek to decrease their costs by forming partnerships with manufacturers to share technologies [67].

### 2.5.2. Processing Time

Blackburn et al. [22] reported that returned products should be managed through a business process and treated as though they had a limited shelf-life; this is because their value is a function of the time elapsed between return and resale. The authors also noted that innovative product categories have short life cycles. Thus, there is a greater reduction in the value of returned products as time passes.

Guide et al. [68] studied a model to maximise value recovery in the product return chain. The researchers suggested a decentralised manufacturing process to improve the efficiency of the return

chain by decreasing the time taken to resell the remanufactured products. Choi et al. [69] used models to demonstrate that retailers should lead product returns because of their proximity to consumers, the greater influence they can exercise, and their interest in speeding the classification and destination labelling processes for returned products. The authors stressed that the longer this process takes, the higher the loss in the product's value; therefore, the greater from the distance between the consumer and retailer, the worse the performance.

Reverse supply chains can be organised according to their use of centralised or decentralised sorting. Blackburn et al. [22] studied the trade-offs between these practices using a model that evaluates the gains in scale from centralised sorting, as against the greater loss in product value because of the longer processing time. Joint marketing strategies between manufacturers and retailers offer benefits to consumers who return products [70].

### 2.5.3. Partnerships with Retailers and Manufacturers

The partnership with retailers is intrinsically related to the processing time variable, as mentioned earlier. From the manufacturer's perspective, the cannibalisation of remanufactured products is a concern. However, the sale of remanufactured products can increase a brand's market share, giving access to consumers who cannot buy new products [71]. The literature explores the benefits of expanding the market to sell remanufactured products instead of cannibalising the sale of new products. It also suggests that remanufactured products should be introduced in the market, even if they reduce the demand for new products [58]. Manufacturers should, thus, take the lead in forming partnerships with retailers and third-party companies to encourage the collection of, and pricing policies for, remanufactured products, as well as take actions that add value [72]. Through policies and partnerships, manufacturers can direct the production of remanufactured products, with the aim of adding value and reducing cannibalisation in line with the product lifecycle [73,74] proposed a model in which the manufacturer directed the behaviour of the consumer through pricing policies, besides offering services, consumer perception, and time of use of the products. Gan et al. [70] concluded that selling remanufactured products using alternative channels, instead of those of retailers, added value to the products because the structure of the channels directed the products to the appropriate group of consumers.

According to Weelden et al. [75], remanufactured product acceptance can be impaired by a lack of familiarity or negative image. Value creation involves offering subsidies for cost–benefit analysis, maintaining good relationships, raising awareness, and generating trust in the products. Guide and Li [76] determined that people reject remanufactured products because of the belief that they are inferior in performance and durability; further, the brand of the manufacturer was a major factor determining the decision to purchase.

Li et al. [77] concluded that manufacturers must hire third-party companies to remanufacture products and add value, especially at the beginning of their life cycle. This would allow them to focus on the development of new products. Decisions regarding remanufacturing are a challenge for manufacturers, who can adopt three strategies: (1) not getting involved; (2) remanufacturing internally; (3) or authorising third-party companies to refurbish their products through partnerships. Not getting involved can lead to lower brand sales, given that some consumers prefer remanufactured and cheaper products. If the manufacturer remanufactures internally, there will be an improvement in the image of the products, which, in turn, would also benefit third-party companies. Thus, the costs of remanufacturing internally need to be analysed and compared with the benefits of partnering with third-party companies [78].

For manufacturers, partnerships with third-party companies facilitate involvement in, and possible control of, the return processes. However, they need to be vigilant to avoid the return of products without any functional or cosmetic defect. Partnerships with retailers increase access to information, tests, and product evaluations, besides helping in the adjustment of consumers' expectations at the purchase point [79]

This article presents a relevant subject, and highlights the factors that affect the value generation by third-party companies in the reverse logistics supply chain of electronic equipment. To the best of our knowledge, there are no articles that contain empirical studies about the impact of these factors in the Brazilian market; this study aims to bridge the research gap.

## 3. Methodology

The research question is 'How do quality, warranty, processing time, and partnerships impact value generation in third-party companies?" The objective is to use case studies to explore quality and warranty, processing time, and partnership in third-party companies, thereby reinforcing the relevance of value generation for these companies. Value generation was analysed by considering the operational results and the vision for the future, expectation of losses or business failure, expectation of keeping the business, or expectation of growth, as reported by the respondents.

The research methodology used was an exploratory qualitative case study. The case study methodology is an appropriate strategy for studying contemporary phenomena in real-life contexts, and is assumed to encompass other contextual conditions related to the cases. Exploratory studies aim to explore situations in which there is no clear range of data to analyse, and identify questions for further research [80].

The units of analysis were third-party companies of reverse electrical and electronic supply chains, defined by convenience. According to Yin [80], it is important to select cases that provide the data required for research. Hence, five companies selling reverse logistic products with different sales and procurement channels were selected. In contrast with the existing literature, the selection of companies selling reverse logistic product is also justified by the peculiar nature of the operation and interfaces.

According to Yin [80], case studies are notable for their capacity to deal with a wide variety of evidence, obtained from documents, interviews, and comments, to describe a phenomenon. Using multiple case studies serves to provide a contrast and the basis to replicate the findings. When analysed separately, each case emphasises the rich context within which the phenomenon occurs. The five companies studied are distinct in terms of size, target market, and processes. Eisenhardt and Graebner [81] recommend the study of 'polar cases', where the researcher can observe the contrasting patterns between different sets of data; this allows him or her to identify the constructs, relations, and logic of the phenomenon studied. As recommended by Eisenhardt and Graebner [81], the choice was not random.

Construct validity was determined through analysis of multiple cases and sources of data. The study planning included mapping the company contacts and defining the cases, interview protocols, visit procedures, and documents needed. Data collection was accomplished using semi-structured interviews conducted by two research participants. The interviewees were owners and senior representatives of companies. A number of subjects were covered, including those related to the mapping of internal processes, interfaces with partners, retailers, buyers, and existing indicators, impressions about difficulties, possibilities for improvement, and market expectations. The interviews were transcribed and analysed at the end of each collection step.

Data analysis was conducted by researchers based on the theoretical framework. The information was organised into Microsoft Office Power Point presentations and shared with the research participants at least twice in order to facilitate the discussion and validation of the results for each company. The results were documented and shared with the interviewees, either in person or by telephone.

According to Yin [80], external validity can be considered as the analytical generalisation. In this article, the factors related to value generation can encompass other scenarios involving third-party companies, and direct their processes to improve overall results. This study summarises the lessons learned from a framework on the impact of quality and warranty, the processing time, and partnerships on the value generation by third-party companies. These three variables were identified in the literature, and the cases were analysed from this perspective, with the aim of relating them to value generation.

The cases representing reverse logistics currently practiced in Brazil for WEEE are illustrated in Figure 1.

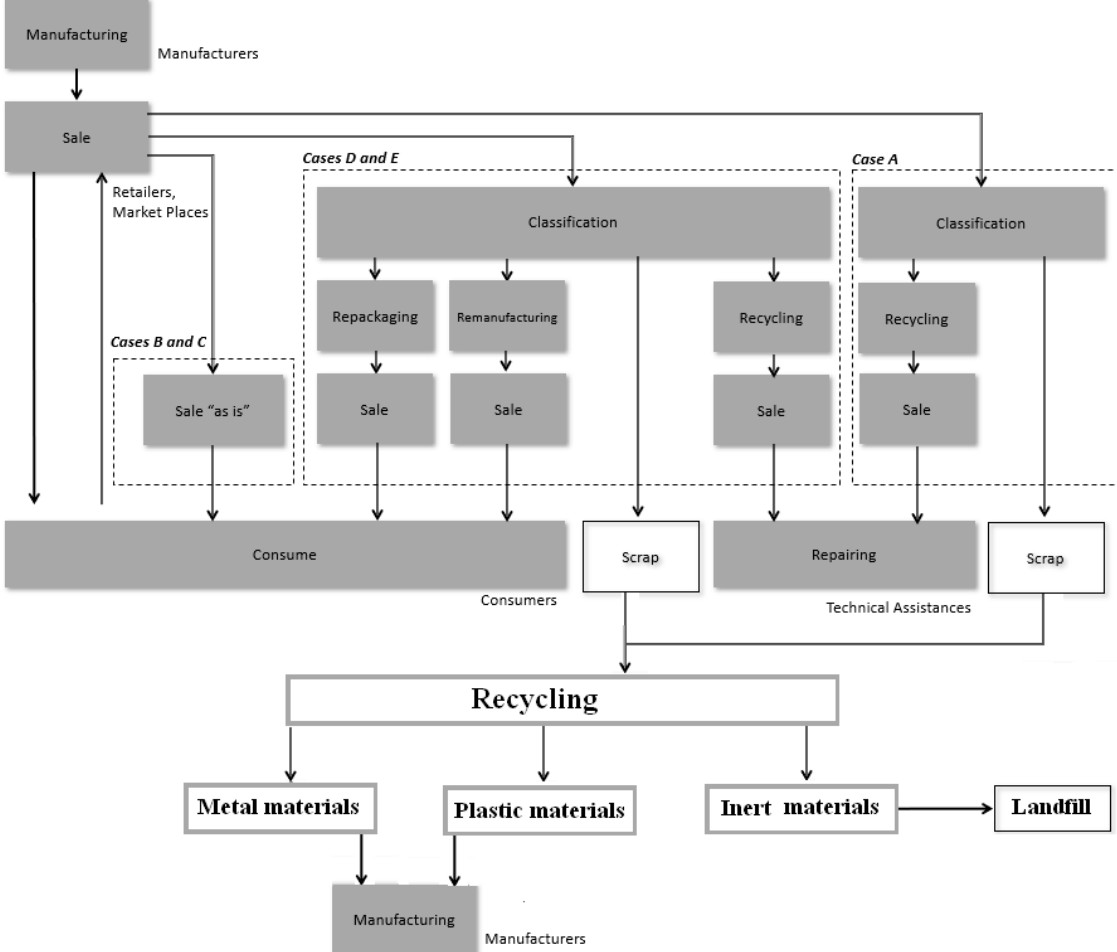

**Figure 1.** WEEE reverse logistics currently practiced in Brazil. Source: Authors.

A conceptual framework and review have been developed by using the methodology of systematic literature review and includes the following aspects [82]:

- formulate research questions;
- select sources for extant literature, such as Scopus, Science Direct, and Web of Science;
- reduce the number of articles by reading them and identifying the main topic;
- define a methodology to analyse the chosen articles;
- describe the main scientific results; and
- identify the scientific gaps and bottle necks.

A bibliographical search of Science Direct, Scopus, and Web of Science was carried out to ascertain the originality of the theme, and find articles that support the discussions and scientifically enrich the text. Using the key word 'electronic waste', a total of 13,332 articles were found. A search using 'electronic waste and logistic' returned 4341 documents; however, only 2076 were scientific articles; of which, we selected those that dealt with reuse, extraction, environmental impacts, legislation, costs, and logistics. Figure 2 shows that research on the subject has increased exponentially in recent years and continues, judging from the articles already published in 2019. The main journal in this field is the Journal of Cleaner Production, followed by Research Resources, Conservation and Recycling, Waste

Management, and the International Journal of Production Economics. Table 1 shows the framework used in this study.

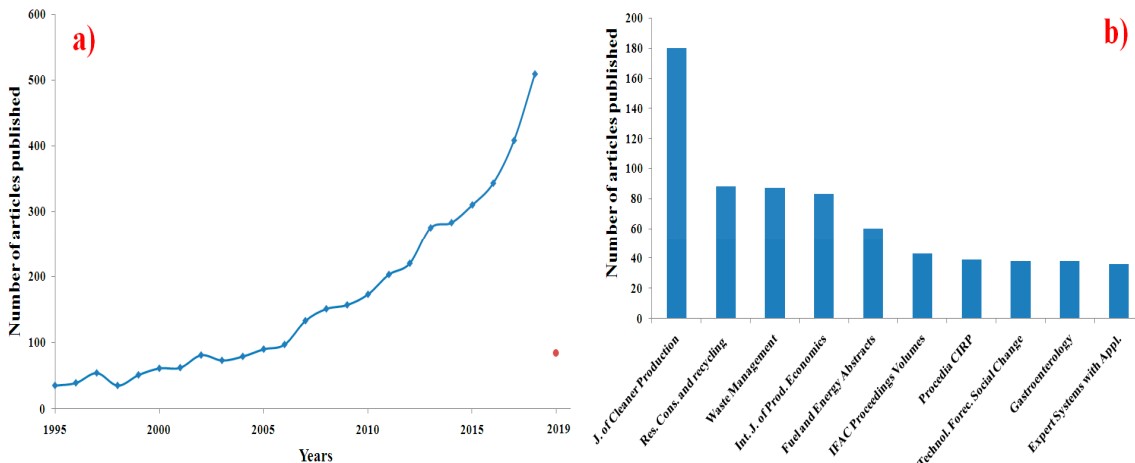

**Figure 2.** Evolution of research on the theme 'WEEE and Logistic'; (**a**) by years and (**b**) by journal. Source: Authors.

**Table 1.** Framework used in this study.

| Themes | Topics Covered | References |
|---|---|---|
| Remanufactured material | They show how remanufacturing lengthens the product life cycle | [5,6,14,21,25,27,28,31,41,51,58,59,64,65, 68,71,73,74,76–78,83–86]. |
| Legislation | They set standards for WEEE in several countries of the world | [23–25,28,29,35,37]. |
| WEEE Logistics | Show the logistics of WEEE-derived materials | [3–5,10,15,22,25,32,37–41,51,53,54,87, 88]. |
| Economic viability | Demonstrate how the processes of extracting and selling recycled materials from the WEEE are economically viable | [2,24,29,55,70,89]. |
| Environmental impacts | Assess the impacts of WEEE on the environment | [1,5,25,29,30,32,33,36,37,47,51,62,63,90]. |
| Health and social impacts | Assess the impacts of WEEE on human health | [1,6,7,25,26,29,30,32,36,37,50,91]. |
| Recycling | Present recycling and reuse processes for WEEE materials | [29,34,40,42–44,46–48,50–52,86,92]. |

## 4. Results and Discussion

### 4.1. Reverse Logistics Sales Companies

#### 4.1.1. Case A

Company A has a large storage area for receiving discarded products, having different levels of failure, and classified into more than 130 categories. Its suppliers are predominantly insurers, while its consumers are either natural or legal persons interested in parts and scraps. The products are sold through electronic auction on 'as is' basis. The company offers potential buyers the option to visit and evaluate the product but does not offer any warranty.

The logistics process consists of the following stages: (1) receiving the products from insurers; (2) classification and registration; (3) negotiation and pricing with the insurer; (4) storage; (5) sorting lots; (6) disclosure of auction; (7) sale; (8) billing; and (9) availability of the product for collection.

### 4.1.2. Case B

Company B has a small storage area, and the process is designed to speed up product receipt, classification, and sale because the physical space available is small. The logistics process involves: (1) retrieving the product from retailers; (2) classification and registration; (3) negotiation and pricing with the retailer; (4) separation and storing lots; (5) disclosure of auction; (6) sale; (7) billing; and (8) availability of the product for collection. The business is focused on electronic appliances, and the sale takes place through a live auction, with the products purchased on 'as is' basis. The weakness of company B is that it relies on a small number of partner suppliers. However, this also differentiates it with respect to the speed of sales; it does not offer a warranty.

### 4.1.3. Case C

Company C manages multi-segment electronic auctions. Given that it is a large company, its organisational structure includes a marketing team that disseminates information about products and auctions. However, because it represents a small portion of the revenue, the focus on more structured partnerships is a limitation. The inventory stays with the retailer, while the products are sold on as is' basis, without any warranty. The total processing time is about 50 days, but can be longer if the minimum agreed value of sales is not realised. In this case, another sales cycle begins, with the aim of renegotiating with interested consumers or new ones. The process consists of: (1) negotiating with the retailer; (2) classification and registration; (3) negotiation and pricing with the retailer; (4) separation of lots; (5) disclosure of auction; (6) sale; (7) billing; (8) scheduling availability for collection; and (9) post-sale.

### 4.1.4. Case D

Company D organises physical sales from a store in a popular neighbourhood. It buys from retailers lots consisting of returned products, inventory surplus, and defective products. The internal structure of remanufacturing lends agility to sale availability. The process consists of: (1) negotiation with retailers; (2) collection; (3) classification; (4) remanufacturing; (5) pricing; (6) sale availability; (7) sale; (8) billing; and (9) availability of the product for collection. Company D offers its own warranty on the operation of its products to consumers.

### 4.1.5. Case E

Company E buys products from retailers in lots consisting of returned products, inventory surplus, and defective products. It has an internal arrangement for reprocessing; it stands out for having partnerships with manufacturers. Through these partnerships, it can offer products that are certified 'reconditioned to factory conditions', in addition to a warranty. The partnership offers credibility to clients, while assuring the manufacturer about the proper destination of its products. The process consists of: (1) negotiation with retailers; (2) collection; (3) classification; (4) remanufacturing; (5) pricing; (6) sale availability; (7) sale; (8) billing; and (9) availability of the product for collection. The availability of products from retailers' accounts for the majority of sales capacity. Thus, company E is structured in a way that maximises its commercial operation.

The cases studies were analysed with respect to their characteristics and the following three variables related to value generation: quality and warranty, processing time, and partnerships with manufacturers and retailers. Only company E is observed to comply with all three proposed variables; this is because the company has signed partnerships with not only retailers, but also manufacturers.

### 4.1.6. Results of Variables That Impact Value-Add in Remanufactured Products

Awareness among the chain members about the potential value and limited life cycle of the returned products is essential; the awareness is mainly about being careful when handling during transportation and storage. Scenario interviewees said that the returned products are treated as scrap,

even before any sorting by retailers occurs. It is, thus, absolutely necessary to train the employees responsible for receiving and classifying the products, at both the retailer and the third-party company. Structured partnerships between manufacturers, retailers, and third-party companies should also consider this difficulty, as it is common in the context of remanufactured products.

Manufacturers can lead partnerships through pricing policies and technology sharing, thereby generating returns, reducing cannibalisation, and preserving the brand image. Research has explored various situations and variables, such as when manufacturers undertake remanufacturing themselves or when they are not involved in returns, wherein partnerships between manufacturers and third-party companies are advantageous. Third-party companies are partners that collect products from retailers, perform remanufacturing, and offer remanufactured products to markets that are distinct from those for new products [21,70–73]. Models have been proposed for evaluating situations in which remanufacturing must be performed by the manufacturer of the product, thereby including remanufactured products in their portfolio. This reduces competition with low-cost products and independent third-party companies [93].

Independent contracting companies are undesirable for manufacturers, primarily because they can compromise the image of the brand when products are resold in channels without the knowledge or authorisation of the manufacturer. Problems with functionality that are treated in a negligent manner can persist long enough to constitute a threat to consumer safety. Remanufacturing by the manufacturer is an alternative considered in the literature; however, the distance between the collection points of retailers and the manufacturers can impair the offering of remanufactured products in the context studied. A decentralised design of remanufacturing and product sale is, therefore, conceived, whereby the remanufacturing operations are shared by the manufacturers and various third-party companies. Establishing partnerships to reduce the activity of independent third-party companies becomes an attractive possibility for manufacturers.

Among the third-party companies studied, company E sought partnerships with manufacturers. Companies B and C have plans to search for partnerships with manufacturers. We observed a tendency among manufacturers to search for partners; this is different from the situation depicted in the literature. With the maturation and growth of the market, these manufacturers are expected to influence channels and sale prices of remanufactured products in return for infrastructure and technology support. Subsidies could be offered for providing adequate remanufacturing support and ensuring the equality and safety of brand products. Manufacturers could also benefit from obtaining information regarding the reason for the return and opinions about the product; the information can be used in new product development, as reported by company E.

During the interviews, those responsible for operations at the contracting companies also discussed their operational results and vision for the future. The presence of all the variables is, thus, related to better results and improved perspectives in the cases studied. We established that the most important factor is the partnership with the suppliers of returned materials, that is, retailers and insurers (in case A); it constitutes the starting point for the operation. After the partnership, the speed of the process is the next variable, keeping in mind the time elapsed between receiving and selling the products. Afterwards, quality and warranty—being related to the additional process of remanufacturing, in addition to the condition of the products received—add value to the products. Finally, the partnership with the manufacturer represents the last variable, as it encompasses fundamental issues regarding the sustainability of the business. These are stated in the literature as follows: consumers of remanufactured products are aware of the origin and warranty of the products, and manufacturers provide information about the returned products and control the correct destination of those products, thereby safeguarding the brand image and reducing cannibalisation. Based on this information, a matrix analysis was created (see Figure 3).

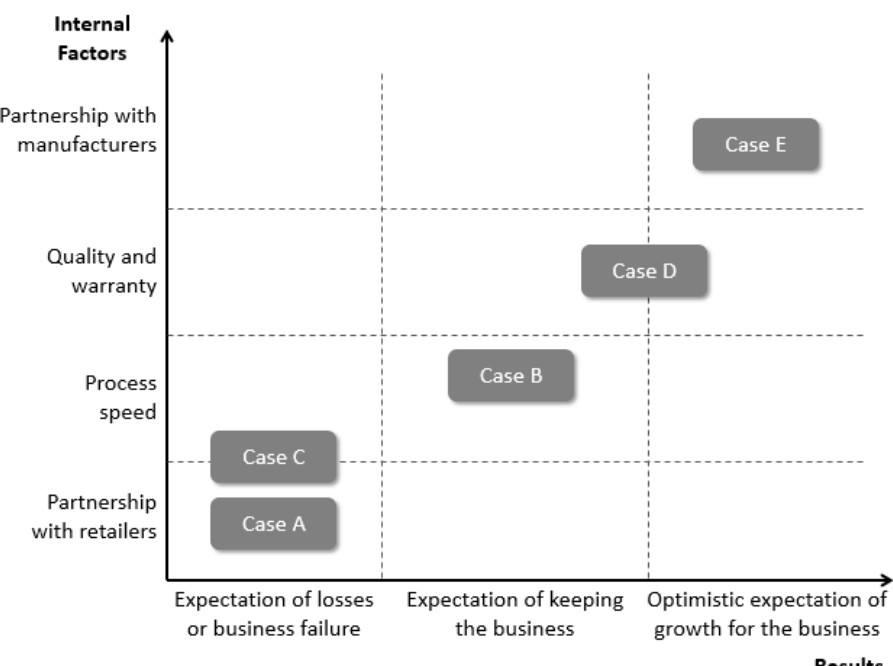

**Figure 3.** Relation between the cumulative presence of internal factors and company outcomes. Source: Authors.

In Figure 3, we can observe the opportunities for generating value from remanufactured products. The process of company E, which has all the variables studied, generates benefits for manufacturers and consumers. Company D has short product buying, remanufacturing, and selling processes and offers its own warranty. It has the opportunity to strengthen its partnerships with manufacturers and improve the image and value-add to its products.

Companies B and C use electronic auctions of products consigned by retailers and do not remanufacture. Thus, they have the advantage of not committing resources to product purchase and remanufacturing. However, their operation focuses on pricing lots and selling the products on 'as is' basis, without warranty, thereby compromising on product value. The lack of manufacturing partners also constitutes a long-term disadvantage because it risks making the company undesirable to the manufacturer, as mentioned.

With respect to processing time, company B transfers the returned products to its inventory and maintains a lean process. Company C maintains its products in retailers' inventory and finalises lot sales after several stages of price negotiation. This could imply that the value of its products is a function of the length of the sales process. For company C, there is an opportunity to add value to the 22 electronic product categories by designing a faster process for certain product categories.

Likewise, company A uses the same process for all 130 categories and could add value by developing specific processes for categories with greater potential. Its operation is characterised by diverse products at different stages of their life cycle, coupled with storage and sale default processes.

Thus, analysing the five companies' situations indicates that the three variables studied could be considered as being related to internal factors for generating value, leading to better results and optimistic expectations for growth. In the interviews, the following external context factors, which could influence third-party companies' results, were reported:

(a) Geographical extent. The distance between manufacturers and retailers increases the freight cost, granting privilege to the operations of third-party companies. The geographical extent contributes to the configuration of a decentralised operation for remanufacturing and selling, wherein the speed of the return chain is emphasised.

(b) Local legislation for warranties and returns. This protects the consumer's right to return products, making it a facilitator for the business model. Despite manufacturers' efforts to increase

knowledge about products and decrease returns, the legal support encourages the operations of third-party companies, ensuring quality raw material in the form of products at the beginning of their life cycle.

(c) Awareness regarding remanufactured products. Consumers, as well as the employees of retailers, manufacturers, and third-party companies, must be aware of the value and utility of remanufactured products. If awareness is present, people tend to be more careful when handling and exhibit a greater sense of urgency when they receive the products, treating them as valuable goods instead of waste.

The interviewees also reported other factors, such as how easy it is to obtain products at the beginning of their life cycle and the life cycle time, that have an influence on their product portfolio. Innovative products have a short life cycle. The impact on their value is, thus, greater in relation to the return time [23]. Therefore, product categories with frequent innovations tend to be avoided, especially in the context of recent market conditions. Furthermore, the risk that portable materials will be stolen may also impair the performance of certain product categories in the context of this study.

The proposed framework summarised in Figure 4 addresses the relation between concepts mentioned in the literature and the observations made in the cases studied: (a) the relationship between the results and the cumulative presence of the variable of value aggregation, characterised by internal factors; (b) the influence of external factors on third-party companies' results; and (c) the factors that influence the product portfolio.

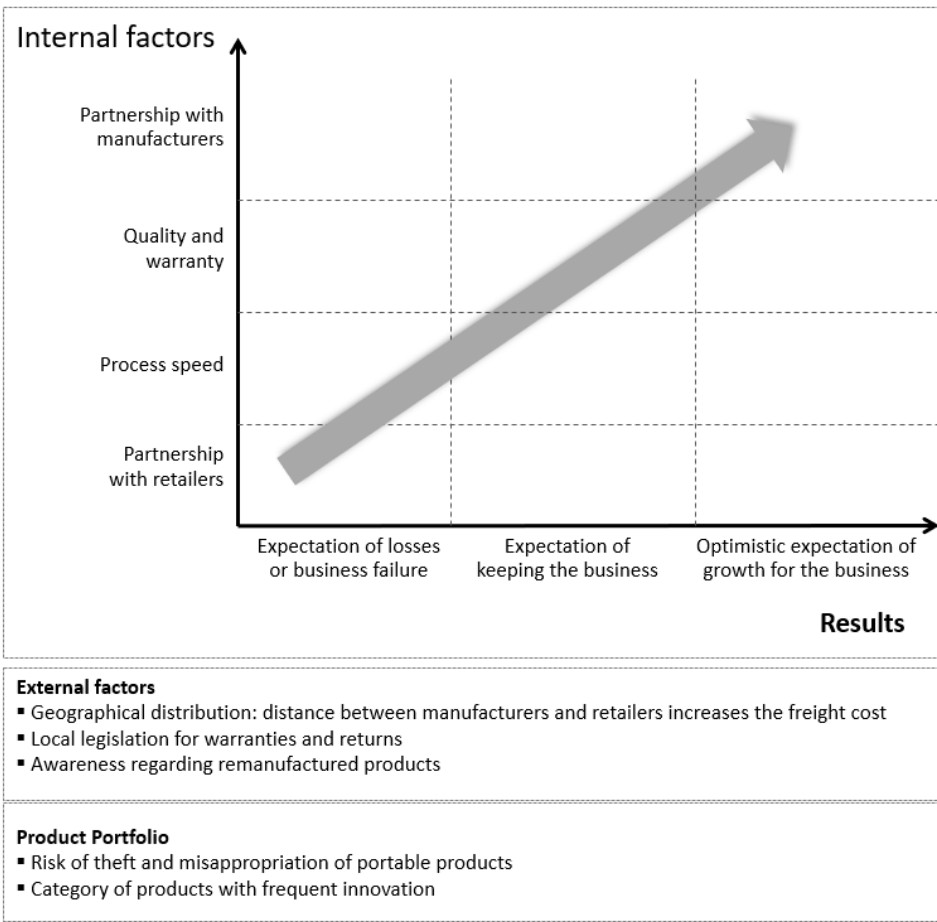

**Figure 4.** Order of relevance of factors found in this research. Source: Authors.

In regard to the Brazilian EEE remanufacturing system, the following advantages can be mentioned as follows:

- Emergence of new companies in the EEE production chain (remanufacturing, logistics, and repair companies);
- With the collection of damaged EEE products and their replacement, the customer portfolio is maintained;
- Increased circulation of capital in the productive chain;
- Reduction of fines for irregular disposal of WEEE;
- Reduction of waste treatment costs because of WEEE reuse and remanufacturing;
- Increase in profits or reduction in losses with the sale of WEEE;
- Improvement in the image of companies;Social
- Increase in the level of employment in the EEE production chain;
- Workers are registered and receive social benefits in accordance with Brazilian labour laws;
- There is no slave-like relationship in the EEA chain in Brazil, as found in other countries;
- As irregular treatments, such as open burning, do not occur, workers are not exposed to toxic substances and there will not suffer diseases caused by dangerous substances present in WEEE;
- There are no employee treatment costs because of diseases caused by WEEE toxic substances;
- Increase in the life cycle of EEE;
- Reduction of WEEE generation;
- With the reuse of EEE parts, there is lower energy consumption;
- Owing to the items mentioned above, there is a lower consumption of raw material for the production of parts and new EEE;
- Treatment of waste according to Brazilian standards;
- Minimisation of WEEE sent to the sanitary landfills;
- As there is no open burning, there is no emission of toxic gases;
- Because of the absence of metals in WEEE intended for landfills, there is no contamination of the soil and groundwater through metal leaching.

In this regard, the legislation and the Brazilian remanufacturing system minimises the impacts caused by spurious relationships in the EEE production chain, which prevents the presence of the philosophy of two worlds beast this Brazilian industrial sector. Disadvantages:

- Owing to the continental dimensions of Brazil, which makes the transportation logistics of WEEE from other regions very expensive, the process only works locally;
- Possible profits from the resale of precious minerals go to other countries;
- There are no companies specialising in the processing of the precious minerals from the WEEE;
- Moving out of specialised jobs related to the processing of precious minerals to other countries.

## 5. Conclusions

Value generation for recycled and remanufactured products can be positive for the stakeholders, such as manufacturers, retailers, third-party companies, and final customers; this makes it a process that should be treated strategically by all the members of the chain. The sales of recycled and remanufactured products are currently being neglected because of product unavailability, unstructured processes, and lack of planning or partnerships, thereby damaging value generation. This study addresses variables related to value generation of reverse logistics for electrical and electronic products.

This article presents the influence of quality and warranty, processing time, and partnerships between third-party companies, manufacturers and retailers, on the value generated from remanufactured products. These variables are related to the best results and optimistic expectations for growth among third-party companies. These internal factors, together with the analysis of external factors and product portfolio, complement the scenario description for the cases studied. The main contribution of this study is to highlight these factors, which are presented in the framework. The lessons learned can be used in other contexts involving third-party companies.

In the US, big retailers grant consumers access to exclusive sessions for remanufactured products. Liquidity Services is one of the major companies in this industry. In Brazil, the sales channels used for remanufactured products are new, and third-party companies are responsible for taking care of a portion of the total demand. Brazil's vast geographical area makes it more expensive to return products to their manufacturers.

The main limitations of this study are related to its methodological choices. More in-depth studies related to external factors, models, and industry players' roles in remanufactured products sales, as well as quantitative indicators for selling remanufactured and recycled products in Brazil, are required. Variables that capture value-add from the perspective of manufacturers and retailers should also be explored.

The adoption of the Brazilian policy for the WEEE has mainly led to the emergence of new companies, an increase in the amount of employment, greater circulation of capital, and lower generation of WEEE; in addition, it has increased the life cycle of products in the EEE productive chain and ensured the treatment of WEEE in accordance with Brazilian environmental law. Brazilian legislation favours EEE remanufacturing and the reuse of WEEE parts in the production chain, reducing as much as possible the shipment of waste to dumping sites and minimising the consumption of energy and the release of toxic substances; these steps are environmentally, socially, and economically correct.

**Author Contributions:** Conceptualization, F.T.B., S.B., A.M.S. and R.M.V.; Methodology, S.B. and A.M.S.; Formal Analysis, S.B. and A.M.S.; Resources, F.T.B. and J.C.C.S.; Writing—Original Draft Preparation, F.T.B., S.B. and A.M.S.; Writing—Review & Editing F.T.B., R.M.V. and J.C.C.S.; Supervision, F.T.B. and J.C.C.S.; Project Administration, F.T.B. and J.C.C.S.

**Acknowledgments:** The authors thank the companies that participated in this research; without their support, this work would not have been possible. We also thank the University of São Paulo (USP), the Nove de Julho University (UNINOVE). This study was financed in part by Conselho Nacional de Desenvolvimento Científico e Tecnológico (CNPQ) and the Coordenação de Aperfeiçoamento de Pessoal de Nível Superior - Brasil (CAPES) - Finance Code 001.

**Conflicts of Interest:** The authors declare no conflict of interest.

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
