# Peer review of "Value Generation of Remanufactured Products: Multi-Case Study of Third-Party Companies"

_sustainability, doi:10.3390/su11030584_

Round 1
Reviewer 1 Report
The paper “Value generation of remanufactured products: multi-case study of third-party companies” deals with an original and interesting topic. Nevertheless, the manuscript presents various limitations that should be improved. Some of the references used to provide an overview of the theoretical background are old-fashioned, therefore they should be updated by recent ones, as explained in the following point. In addition, considering the introduction, and particularly concerning the crucial role of the impact of sustainability in 3PLs, it is not possible to neglect the role of the emerging green practices and the emerging technological tools. Marchet et al., and Centobelli et al., 2017 proposed respectively a definition of green initiative and a WH2 framework that takes into account if the environmental initiatives are external or internal to the organization. I believe that your work should also consider that work for discussing and justifying the motivation of your perspectives of analysis. As for methodology, a short flowchart summarizing the methodological steps will help the readers to understand the methodology. Furthermore, which are the strength points and weakness points of this methodological approach? In addition, the authors should describe and justify more in details the choice of multiple case studies, and how their application converge or diverge from other similar studies provided in the literature. Moreover, the authors should describe more in details if and how their application converge or diverge from the standard within-case and cross-case analysis methods. Finally, the authors should stress the implications for policy makers and the limitations of this study?
Author Response
Respose to reviewer
In the new version we have inserted several references of the most recent dates, including green practices in the WEEE field.
New items have been inserted that address the environmental and social problems of WEEE, as well as the recycling techniques of WEEE-derived materials
At the end of the methodology we have inserted a step by step of the research methodology, including an updated bibliographic review framework
At the end of the discussions we inserted the advantage because to the adoption of reverse logistics in EEE remanufacturing companies. We also insert disadvantage. However, the limitations of the research were inserted in the conclusions.
We thank the reviewer for suggestions that will improve the quality of our article
Reviewer 2 Report
The paper “Value generation of remanufactured products: multi-case study of third-party companies” introduces the influence of quality and warranty, processing time and partnerships between thirdparty companies, manufacturers and retailers, on the value generation of remanufactured products. The content of the paper is adequate for the purposes of the journal.
Title: The title of the paper is informative. It includes important terms and the message of the article.
Abstract: The abstract describes the context and follow the structure: backgrounds, methods, results and conclusions.
Keywords: Keywords are well chosen.
Introduction. Introduction defines the focus and the research questions. It does not explain the structure of the text. The sustainability aspect of the article must be also explained in this part of the text.
Literature review: However the literature review supports to understand the correlation of presented research results with literature but I suggest adding more references to enhance the literature review, especially from the green supply chain point of view (e.g. see in DOI:10.3390/en11071833). A summary table comparing the contributions could support the explanation.
Methodology. The methodology is discussed, but I suggest increasing the scientific soundness of the description of methodology. The quality (resolution) of Figure 1 must be improved.
Results and discussion. In my opinion, this is the weakest part of the paper. I suggest adding more analysis to increase the scientific soundness of the article.
Style. Please, use mdpi template!
Author Response
Response to reviewer
In the new version we have inserted several references of the most recent dates, including green practices in the WEEE field. New items have been inserted that address the environmental and social problems of WEEE, as well as the recycling techniques of WEEE-derived materials
At the end of the methodology we have inserted a step by step of the research methodology, including an updated bibliographic review framework; including the article indicated by the reviewer.
At the end of the discussions we inserted the advantage because to the adoption of reverse logistics in EEE remanufacturing companies. We also insert disadvantage.
We thank the reviewer for suggestions that will improve the quality of our article
Round 2
Reviewer 2 Report
Please, check the language and typing again!
Author Response
Dear reviewer’s, good afternoon!
Please see the revised manuscript, I wold like to inform you that the article has been revised by Elsevier Language Editing.
Best regards,